# The Evolution of Blood Cell Phenotypes, Intracellular and Plasma Cytokines and Morphological Changes in Critically Ill COVID-19 Patients

**DOI:** 10.3390/biomedicines10050934

**Published:** 2022-04-19

**Authors:** Elisabeth Berghäll, Michael Hultström, Robert Frithiof, Miklos Lipcsey, Victoria Hahn-Strömberg

**Affiliations:** 1Department of Surgical Sciences, Anesthesiology and Intensive Care, Uppsala University, 75185 Uppsala, Sweden; michael.hultstrom@mcb.uu.se (M.H.); robert.frithiof@surgsci.uu.se (R.F.); miklos.lipcsey@surgsci.uu.se (M.L.); 2Department of Medical Cell Biology, Uppsala University, 75123 Uppsala, Sweden; victoria.stromberg@mcb.uu.se; 3Hedenstierna Laboratory, CIRRUS, Anesthesiology and Intensive Care, Department of Surgical Sciences, Uppsala University, 75185 Uppsala, Sweden

**Keywords:** severe COVID-19, flow cytometry, immunophenotype, morphology, intensive care

## Abstract

Background: Severe coronavirus disease 2019 (COVID-19) causes a strong inflammatory response. To obtain an overview of inflammatory mediators and effector cells, we studied 25 intensive-care-unit patients during the timeframe after off-label chloroquine treatment and before an introduction of immunomodulatory drugs. Material and methods: Blood samples were weekly examined with flow cytometry (FCM) for surface and intracytoplasmic markers, cytokine assays were analyzed for circulating interleukins (ILs), and blood smears were evaluated for morphological changes. Samples from healthy volunteers were used for comparison. Organ function data and 30-day mortality were obtained from medical records. Results: Compared to that of the healthy control group, the expression levels of leukocyte surface markers, i.e., the cluster of differentiation (CD) markers CD2, CD4, CD8, CD158d, CD25, CD127, and CD19, were lower (*p* < 0.001), while those of leukocytes expressing CD33 were increased (*p* < 0.05). An aberrant expression of CD158d on granulocytes was found on parts of the granulocyte population. The expression levels of intracellular tumor necrosis factor alpha (TNFα) and IL-1 receptor type 2 in leukocytes were lower (*p* < 0.001), and the plasma levels of TNFα, IL-2, IL-6, IL-8, IL-10 (*p* < 0.001), interferon gamma (IFNγ) (*p* < 0.01), and granulocyte-macrophage colony-stimulating factor (GM-CSF) (*p* < 0.05) were higher in patients with severe COVID-19 than in the control group. The expression levels of CD33+ leukocytes and circulating IL-6 were higher (*p* < 0.05) among patients with arterial oxygen partial pressure-to-fractional inspired oxygen (PaO_2_/FiO_2_) ratios below 13.3 kPa compared to in the remaining patients. The expression levels of TNFα, IL-2, IL-4, IL-6, IL-8, and IL-10 were higher in patients treated with continuous renal replacement therapy (CRRT) (*p* < 0.05), and the levels of the maximum plasma creatinine and TNFα Spearman’s rank-order correlation coefficient (rho = 0.51, *p* < 0.05) and IL-8 (rho = 0.44, *p* < 0.05) correlated. Blood smears revealed neutrophil dysplasia with pseudo-Pelger forms being most common. Conclusion: These findings suggest that patients with severe COVID-19, in addition to augmented ILs, lymphopenia, and increased granulocytes, also had effects on the bone marrow.

## 1. Introduction

The spread of severe acute respiratory syndrome coronavirus 2 (SARS-CoV-2) results in the coronavirus disease 2019 (COVID-19) pandemic ranging from asymptomatic to acute respiratory distress syndromes (ARDSs), multiorgan failure, and death [1]. Moreover, acute kidney injury (AKI) and intensive care unit-acquired critical illness weakness (ICUAW) are frequent complications in critically ill patients [2,3].

The broad panorama of illness severity probably represents the variations in the complex immune response to the SARS-CoV-2 infection. The acute pro-inflammatory interleukin (IL) response with IL-1, IL-6, and tumor necrosis factor alpha (TNFα) levels increases and causes immune cell infiltration in the affected tissues [4]. Immune effector cells and inflammatory, complement plus coagulation plasma proteins are directed to attack extracellular microorganisms, as well as infected and uninfected cells [5,6,7,8].

Apart from the local inflammation, the activation of circulating immune cells has been suggested to be of pathophysiologic and prognostic value. Early in the COVID-19 pandemic, a poor outcome was associated with a dysfunctional adaptive T cell response, a low total lymphocyte count, decreased levels of T-helper (TH), and cytotoxic T lymphocytes (CTL) [8,9]. Later, studies confirmed this and added natural killer cell (NKC) cytopenia plus the decreased intensity of Human Leukocyte Antigen DR isotype (HLA-DR) on cluster of differentiation 14 positive (CD14+) monocytes [10,11]. Another study found a B-cell response including loss of germinal center formation and functional follicular TH cells in patients with severe disease compared to patients with mild disease or healthy controls (HCs) [12]. The inflammatory effect caused by mastcells (MCs) is prominent in cytokine storm caused by COVID-19. In addition, the prevalence of mastcell activation syndrome (MCAS) of up to 17% is similar to the incidence of severe disease from COVID-19. Drugs against MC-mediated effects have preliminary effects against COVID-19 symptoms, although no patients with severe COVID-19 were treated. Over-activated MCs are an important factor in the development of fibrotic conditions [12,13,14]. Activated neutrophils form neutrophil extracellular traps (NETs) that attack invading microorganisms in blood and induce a pro-coagulative state [15]. Moreover, membrane markers for complement regulation such as CD55 and CD59 plus markers for intracellular levels of cytokines serve as markers for the immune systems’ direction and capacity [16,17,18,19].

Many studies have correlated COVID-19 severity with clinical, radiological, physiological, and laboratory parameters including the complete blood count with lymphocytopenia, neutrophilia, eosinophilopenia, mild thrombocytopenia, and circulating cytokine levels, but few have focused on blood cell phenotypes, intracellular cytokines, and morphological changes over time and their relation to illness severity and organ failure [20,21].

As different immunological effector cells characterized by surface markers and plasma ILs are likely to contribute to illness severity, morbidity, and mortality from COVID-19, we hypothesized that changes in circulating immune cells and mediators are of importance for outcome in a critically ill COVID-19 cohort.

In this longitudinal prospective observational study, we set out to describe possible key immunological markers and hematomorphological changes associated with severe COVID-19, mortality, and organ failure in blood from critically ill COVID-19 patients.

## 2. Materials and Methods

A longitudinal prospective observational study was performed in critically ill patients with COVID-19 included in the Pronmed-study. The study was approved by the Swedish Ethical Review Authority Dnr 2017/043 (with amendments of 2020-01623, 2020-02719, 2020-05730, 2021-01469, and 2022-00526-01). Patients at >18 years of age, with reverse-transcription polymerase chain reaction (RT-PCR)-confirmed COVID-19 on nasopharyngeal swabs and treated in the intensive care unit (ICU) of the Uppsala University Hospital, Sweden, were recruited and followed for 30 days. Informed consent was obtained from the patient or next of kin if the patient was unable to give consent.

Patients admitted to the ICU between 13 April and 23 June in 2020 were all treated with assisted ventilation, non-invasive or invasive, according to the standard care. During this period, chloroquine treatment had ceased, and advanced immunomodulatory treatment had not yet been introduced. Examinations were performed on clinical indication. Inclusion criteria for the subpopulation examined with flow cytometry (FCM) and Luminex were being hospitalized in the central subunit of our ICU, with access to more advanced respiratory treatment plus continuous renal replacement therapy (CRRT), at noon the day before sample collection. All received invasive ventilation. Exclusion criteria were having admitted to our temporary COVID-19 ICU subunit or being admitted to our central ICU subunit, but not at noon the day before sample collection. The number of patients was limited by FCM analysis capacity. The study flow diagram is shown in Appendix A.

Control samples were collected from healthy individuals recruited among health care workers, approved by the Swedish Ethical Review Authority Dnr 2020–03056. The Declaration of Helsinki with revisions was followed, and STROBE guidelines for cohort studies were used for reporting (https://www.strobe-statement.org/checklists/ accessed on 25 March 2022).

### 2.1. Clinical Variables

The low arterial oxygen partial pressure-to-fractional inspired oxygen ratio (PaO_2_/FiO_2_) was defined as <13.3 kPa. CRRT was started according to the discretion of the attending clinician. ICUAW was defined as neurophysiologic investigations showing signs of critical illness polyneuropathy or myopathy. Thirty-day mortality was registered. Blood biochemistry was analyzed by the hospital’s central laboratory.

### 2.2. FCM

Once a week, peripheral blood was collected in a 5 mL EDTA tube for FCM and cytokine analysis until death or ICU discharge. Immunophenotype characterization of leukocytes was defined as CD45+ using FCM. Antibodies for surface and intracytoplasmic markers were obtained from Miltenyi Biotec (Appendix A). Fifty–one hundred microliters of blood were added to each tube within an hour from sample harvesting and prepared according to the manufacturers’ protocol. Briefly, tubes were prepared with an antibody, incubated and washed before erythrocyte lysis. Then, the washed cell pellets were resolved in a fixation buffer. For the detection of intracellular antigens, a higher concentration of a lysing buffer was used. As a negative control, blood without an antibody marker and Ig isotype controls was used, and the example is shown in Appendix A. Samples with surface markers were analyzed within six hours, and samples with intracytoplasmic markers were examined within three hours in a flow cytometer (NAVIOS, Beckman Coulter, Inc. Brea, CA, USA). Software analysis was conducted with Kaluza Issue AC (version 2.1).

### 2.3. Multiplex Cytokine Analysis

A 5 ml EDTA tube was centrifuged at 1300× *g* for 10 min. Plasma aliquots were stored in Eppendorf tubes at −20 °C for later characterization of circulating ILs. Levels of IL-2, -4, -6, -8, and -10, TNFα, granulocyte-macrophage colony-stimulating factor (GM-CSF), and interferon gamma (IFNγ) were analyzed in plasma at a dilution rate of four, with a Bio-Plex Human Cytokine Assay using a Luminex MagPix instrument (Bio-Rad Laboratories AB, Sundbyberg, Sweden). The parameters of cytokine are shown as followings: for IL-2: limit of detection (LOD), 0.75 pg/mL, intra-assay coefficient of variance (CV), 1.7%, inter-assay CV, 2.5%; IL-4: LOD, 0.09 pg/mL, intra-assay CV, 3.2%, inter-assay CV, 1.9%; IL-6: LOD, 0.34 pg/mL, intra-assay CV, 2.2%, inter-assay, 3.0%; IL-8: LOD, 0.36 pg/mL, intra-assay CV, 3.2%, inter-assay CV, 2.8%; IL-10: LOD, 0.69 pg/mL, intra-assay CV, 2.3%, inter-assay CV, 3.4%; TNFα: LOD, 1.13 pg/mL, intra-assay CV, 3.5%, inter-assay CV, 3.0%; GM-CSF: LOD, 0.19 pg/mL, intra-assay CV, 4.3%, inter-assay CV, 2.2%; IFNγ: LOD, 1.05 pg/mL, intra-assay CV, 3.1%, inter-assay CV, 3.6%.

### 2.4. Morphology

Blood smears were weekly collected from 16 patients and HCs (12), stained with May–Grünwald–Giemsa and prepared for morphology. The findings of dysplasia, anisocytosis, as well as progenitor cells were noted. Dysplasia was graded semi-quantitatively into four categories of abnormal nucleus, i.e., pseudo-Pelger, hypogranulation, hypergranulation, and hypersegmentation. Morphologic assessment was added to the protocol when aberrant expressions were found with FCM to examine effects on bone marrow; thus, the number of patient samples was lower than for FCM. Analysis was made on anonymized slides by an experienced morphologist (author: Victoria Hahn Strömberg).

### 2.5. Statistics

Continuous variables are presented as mean (SD) or median (interquartile range (IQR)), and the count variable was the number (percent of the total number of observations). For group comparisons, the Mann–Whitney U test was used. The Spearman’s rank-order correlation coefficient was used to assess associations. Cytokines with concentrations below the detection level were given as the value half of the LOD. Statistica software (version 13.5; TIBCO Software Inc., Tulsa, OK, USA) was used.

## 3. Results

Of 78 patients admitted to the ICU during the time of the study, weekly peripheral blood FCM and plasma IL analyses were performed on 25 patients, and blood smear morphology was performed on 16 patients. The characteristics of the total ICU cohort and the FCM patients are presented in Table 1. Three patients died. The duration of self-reported illness before admission to the ICU varied between two and 25 days, with a median of 9 (8–12) days. The 12 controls included in the study were healthy individuals, including five women and seven men. Their mean age were 39 years (range: 31 to 49 years).

### 3.1. Surface and Intracytoplasmic Leukocyte Markers

Patients with COVID-19 showed lower expression levels of lymphocyte surface CD markers during the first week for all the examined T subsets (CD2, CD4, CD8, CD158d, and CD25) as well as CD127 and B-cell marker CD19 compared to HCs (*p* < 0.001 for all; Table 2). The levels of leukocytes expressing CD33 were increased compared with in HCs (*p* < 0.05). For HLA-DR, CD203c, and complement markers CD55 and CD59, there were no differences between the groups. The levels of intracellular TNFα and interleukin 1 receptor type 2 (cIL1R2) in leukocytes were higher in HCs than in COVID-19 patients during their first ICU week (*p* < 0.001; Table 2). Due to few values obtained in weeks 2–6, statistical analysis was only performed for the first ICU week. The decrease of intracellular cytokines was previously described and may represent leukocyte exhaustion [19]. Although few values were measured for weeks 2–6, the levels seemed to increase over time in patients that were still admitted to the ICU. No differences were seen for intracellular IL-6, IL-8, and IL-10 between COVID-19 patients and HCs.

### 3.2. Aberrant Expression of CD Markers

In samples from two patients, an aberrant expression of CD158d in the granulocyte population was found. There was a tendency for the aberrant expression to diminish over time. One coexisting aberrant expression of CD2, CD19, and CD203c was observed. No other aberrant expressions were found (Figure 1).

### 3.3. Circulating Cytokines

Plasma levels of TNFα, IL-2, IL-6, IL-8, IL-10 (*p* < 0.001), IFNγ (*p* < 0.01), and GM-CSF (*p* < 0.05) were higher in COVID-19 patients compared to in HCs. No difference was found for IL-4 (Figure 2).

### 3.4. FCM Markers and Circulating Cytokine Levels Related to Organ Failure and Mortality

There was no correlation between the measured surface markers, intracellular cytokines and PaO_2_/FiO_2_ ratio. However, for patients with PaO_2_/FiO_2_ ratios less than 13.3 kPa vs. patients with better respiratory function and PaO_2_/FiO_2_ ratios of ≥13.3 kPa, the myeloid marker CD33 (89.9 ± 7.8% vs. 84.6 ± 2.2%) and plasma IL-6 levels (234.0 ± 518.9 vs. 2.3 ± 1.7 pg/mL; *p* < 0.05) were higher in the former. 

Levels of IL-2, IL-4, IL-6, IL-8, IL-10, and TNFα in the first ICU week were higher in patients who were treated with CRRT during their ICU admission (*p* < 0.05). The levels of the maximum plasma creatinine and circulating TNFα Spearman’s rank-order correlation coefficient (rho = 0.51, *p* < 0.05) and IL-8 (rho = 0.44, *p* < 0.05) correlated (Table 3). 

CD14 expression was lower during the first ICU week among patients who developed ICUAW (*p* < 0.05). CD14 expression was not associated with the length of stay in the ICU. There were no differences in surface CD markers and circulating cytokine levels between patients who survived for 30 days after ICU admission and non-survivors.

### 3.5. Morphology

Morphological dysplasia within the neutrophil granulocyte population was notable with pseudo-Pelger forms in all 16 patients as well as hypersegmentation in 12 patients, hypogranulation in 10 patients, and hypergranulation in 10 patients. Thrombocyte anisocystosis was seen in all samples, and erythrocyte anisoscystosis was observed in half of the patients. Progenitor cells were found in 10 of the 16 patients, predominantly within the myeloid lineage with promyelocytes and myelocytes being the most common. Orthochromatic erythroblasts were seen in six of the patients, and blast cells were observed in four. The results from the first-ICU-week blood smears are shown in Table 4. There was generally no difference regarding dysplasia between early and late samples obtained from the same patients in our data, except for one patient who had no dysplasia during the last week. Nine of the patients were followed for two weeks or more, and two patients were followed for six weeks. Within the HC group, three samples showed platelet anisocystosis, and one showed erythrocyte anisocystosis. No dysplasia in the form of pseudo-Pelger forms or progenitor cells was found in any cell lineage among the control samples.

## 4. Discussion

In this prospective study of critically ill COVID-19 patients, the lymphocyte expression levels of CD markers for the examined T cell subsets (CD2, CD4, CD8, CD158d, and CD25) and CD127 as well as CD19 were lower in COVID-19 patients than in HCs. An aberrant expression of CD158d in parts of the granulocyte population was observed in two patients. Leukocytes levels expressing the myeloid marker CD33 were increased in COVID-19 patients. Moreover, CD33 and circulating IL-6 levels were higher among patients with severe respiratory failure, i.e., PaO_2_/FiO_2_ ratio of <13.3 kPa. Levels of IL-2, IL-4, IL-6, IL-8, IL-10, and TNFα were higher in patients who were treated with CRRT (*p* < 0.05). The maximum plasma creatinine and circulating TNFα and IL-8 levels correlated. All morphologically examined patients showed dysplasia within the neutrophil granulocyte population.

The lower expression levels of lymphocyte surface CD markers for T cell subsets and B-cells are consistent with previous findings, as well as the increased levels of circulating cytokines and their relation to organ failure [2,8,22,23].

The aberrant expression of CD158d for the killer cell immunoglobulin-like receptor KIR2DL4 observed in parts of the granulocyte populations in two patients has not been described in severe COVID-19 before, and the significance is yet to be explained. CD158d binds to MHC I subset HLA-G and inhibits or activates NKCs, depending on the intracellular receptor part. The CD158d gene show large genetic polymorphism within the human population and between species. HLA-G expression is induced in pregnancy and some autoimmune, malignant, and infectious diseases [24,25,26]. In an early case report from a recovering COVID-19 patient, a high-low-high HLA-G variation in lymphocyte and monocyte population with stable levels of CD158d was described [27]. Induced immunosuppression by HLA-G and its receptors, for example KIR2DL4, in SARS-CoV-2 was further discussed in Frontiers in Immunology [28]. For samples with an aberrant expression of CD158d, some of the highest levels of IL-4, IL-6, IL-8, TNFα, and GM-CSF were observed. The cytokine levels were, in general, above mean values, although the GM-CSF level decreased below the detection level over time. The presence of an aberrant expression in two separate patients, the observation that it faded with clinical recovery plus the combination of high cytokine levels, suggest this may be of relevance for disease severity or recovery. Both patients survived.

Our study showed decreased levels of CD127, a lymphocyte receptor for IL-7, during the first ICU week. Circulating IL-7 is essential for lymphocyte proliferation and terminal differentiation, and its levels were shown to be higher in patients with severe COVID-19 than those with non-severe COVID-19 [29]. A preprint found an expansion of CD127 expressing lymphocytes associated with milder disease [30]. IL-7 substitution is a potential treatment to restore lymphocyte count and function and reduce mortality in severe COVID-19 [31]. A decrease in CD127 expression in severe COVID-19 patients may be a significant factor in IL-7 treatment.

A decreased amount of CD14 during the first ICU week was found in a group of eight patients who developed ICUAW, and a trend towards increasing values over time could be seen. Five of them were treated for more than a month. A previous study characterized mononuclear cell populations in peripheral blood in the early phase of COVID-19 non-survivors compared to in survivors after 28 days. They found a monocyte subpopulation with reduced CD14 and HLA-DR expression, despite the preserved total monocyte count and an increased CD4/CD8 ratio in favor of a TH2-like lymphocyte phenotype [32]. Another study found that an immunological profile towards TH2 and M2-type macrophages were an independent risk factor for death [11]. It is possible that a low expression of CD14 can be an early predictor for long ICU care, ICUAW, or death in a larger cohort. However, in our study, there was no association between CD14 and the length of stay in the ICU.

The observed decrease in intracellular TNFα and IL1R2 levels during the first week tended to recover over time in patients with a long length of stay in our cohort. Decreased levels of intracellular IL-2, IFNγ, and TNFα were previously reported in patients with severe disease compared to those with mild disease [19].

Morphological findings revealed morphological dysplasia in all hematopoietic linages. Signs of dysplasia were notable in the myeloid lineage where all patients had pseudo-Pelger forms, and some showed hypogranulation, hypergranulation, and/or hypersegmentation in addition to the pseudo-Pelger forms. In addition, several patients showed progenitor cells within the myeloid and the erythroid lineages in the blood smears. This is probably caused either by SARS-CoV-2, the immune response, or both. Several patients showed an ordinary morphology by the time they were discharged from the ICU. The abnormalities can be related to hyperinflammation caused by cytokine storm in COVID-19 and may be a form of secondary hemophagocytic lymphohistiocytosis [21]. Two patients were diagnosed with hematologic malignancies prior to admission or within 90 days. Higher mortality has been described in hematological than solid malignancies among COVID-19 patients, and low CD8+ T-cells was associated the highest mortality rate [33]. Whether this is due to impaired immune cell function or the infection contributes to the deterioration of hematological malignancies needs to be further evaluated.

A strength of the study is that it was unaffected by therapeutic anti-inflammatory interventions, since it was conducted after treatment with chloroquine ceased and before treatment with steroids and other immunomodulatory drugs were introduced. The number of virus variants due to mutation was limited during this period. Another strength is the prospective design enabling standardized sample handling and measurements as well as structured data collection. Limitations are that patients with severe COVID-19 who were not treated in the ICU were not included and a comparison of these patients with a group of patients with mild disease would have increased our possibility to discriminate between key parameters for milder and severe diseases. FCM is a time-sensitive and labor-intensive method, which limited the number of possible samples. Individual results would have been interesting to discuss, although this is not a part of our ethical permission. The resulting sample was representative of the population as a whole (Table 1). However, the difference in respiratory treatment and minor differences in the co-morbidity between the populations suggests that our subpopulation may have had a greater respiratory impairment and require invasive respiratory treatment to a greater extent. Some patients in the entire population may have been assessed as too fragile to benefit from invasive respiratory treatment. As an observational study, without intervention, the results should not be interpreted as the cause of disease severity or mortality. Since the material is small and all patients were severely ill, the subgroup analysis regarding organ failure and mortality should be interpreted with caution. 

In conclusion, patients with severe COVID-19 showed a leukocyte profile with predominance for myeloid markers and lower-surface markers for lymphocytes including CD127, a receptor for IL-7. The expression may be important in treatment with pharmacological IL-7, a potential drug for severe disease that has been successful in restoring lymphocyte count and function in patients with non-severe COVID-19 [30]. An aberrant expression of CD158d in parts of the granulocyte population was seen as well as the expression of CD2 and CD19 within the granulocyte population, indicating that SARS-CoV2 also effects the bone marrow, resulting in not only phenotypic changes, but also the dysplasia of cells and cell complexes within the myeloid lineage.

## Figures and Tables

**Figure 1 biomedicines-10-00934-f001:**
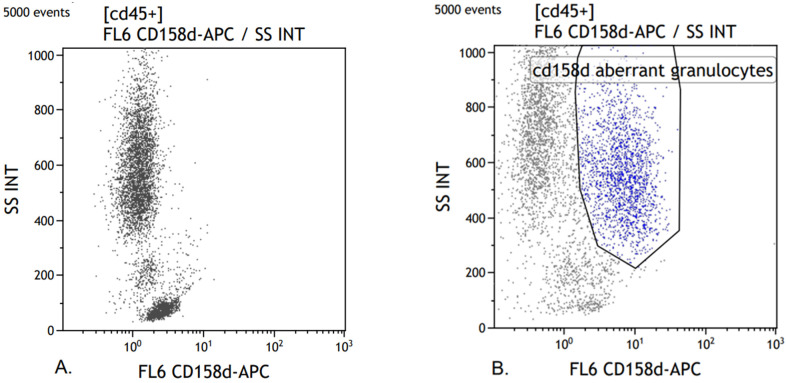
FCM scatterplots for CD158d: (**A**) sample from HCs; (**B**) sample from a patient with an aberrant expression of CD158d in the granulocyte population, with CD158d+ granulocytes gated.

**Figure 2 biomedicines-10-00934-f002:**
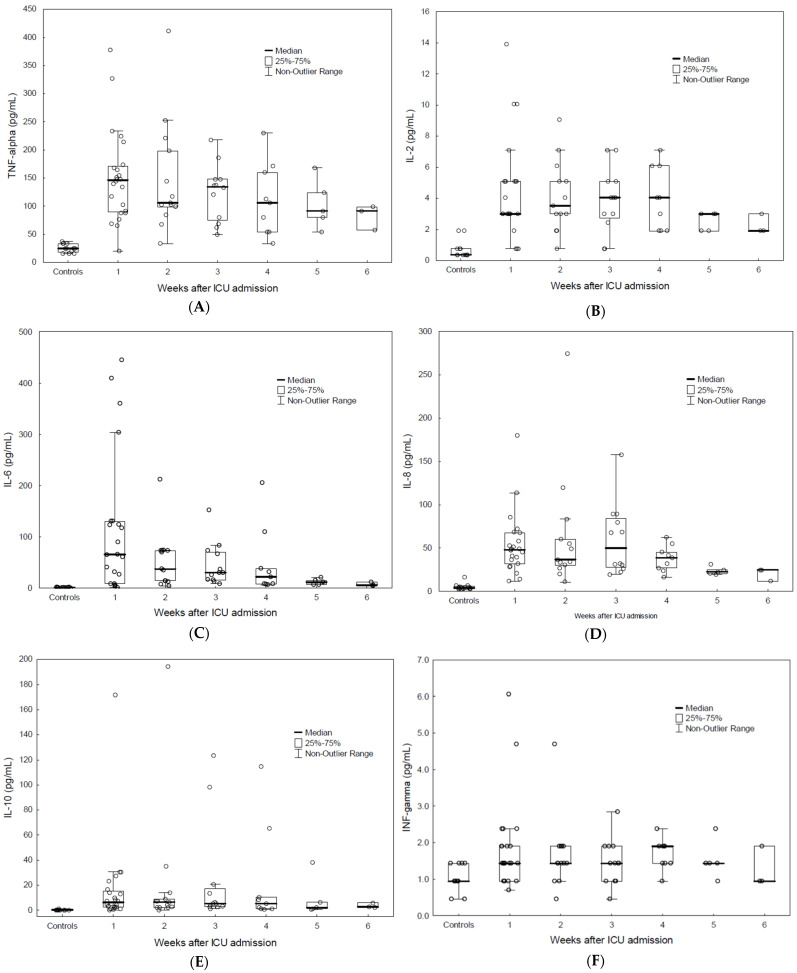
Weekly plasma cytokine levels in ICU patients with severe COVID-19 compared to in HCs. Levels of TNFα, IL-2, -6, -8, and -10 (*p* < 0.001), IFN-γ (*p* < 0.01), and GM-CSF (*p* < 0.05) were higher in COVID-19 patients compared to in HCs: (**A**) TNFα-levels; (**B**) IL-2 levels; (**C**) IL-6 levels; (**D**) IL-8 levels; (**E**) IL-10 levels; (**F**) IFNγ levels; (**G**) GM-CSF levels. Due to a few values for weeks 2–6, statistical analysis was only performed for the first ICU week.

**Table 1 biomedicines-10-00934-t001:** Demographic and clinical characteristics.

	All Patients	Flow Cytometry (FCM) Patients
	*n* = 78	*n* = 25
Age (years: mean ± SD)	61 ± 13	60 ± 13
Weight (kg: mean ± SD)	90 ± 24	92 ± 26
BMI (kg/m^2^: mean ± SD)	30 ± 7	31 ± 8
Female (*n*, %)	20 (38)	6 (24)
SAPS3 (mean ± SD)	53 ± 10	53 ± 9
LPC (10^9^/L: mean ± SD)		8.5 ± 4.1
Comorbidities		
Chronic pulmonary disease (*n*, %)	21 (27)	6 (24)
Hypertension (*n*, %)	37 (47)	11 (44)
Heart failure (*n*, %)	4 (5)	2 (8)
Ischemic heart disease (*n*, %)	7 (9)	3 (12)
Previous thromboembolic event (*n*, %)	7 (9)	2 (8)
Malignancy (*n*, %)	6 (8)	2 (8)
Diabetes mellitus (*n*, %)	19 (24)	4 (16)
Neurologic disease (*n*, %)	3 (4)	3 (12)
Non-smoker (*n*, %)	59 (77)	18 (72)
Medications prior to admission		
Steroid treatment (*n*, %)	7 (9) ^1^	5 (20) ^1^
ACEi or ARB treatment (*n*, %)	23 (30)	9 (36)
Anticoagulant treatment (*n*, %)	17 (22)	4 (16)
Mortality and organ failure in the ICU		
Mortality of 30 days (*n*, %)	10 (13)	3 (12)
Mortality of 90 days (*n*, %)	15 (19)	4 (16)
PaO_2_/FiO_2_ ratio of <13.3 kPa (*n*, %)	44 (82) ^2^	16 (89) ^2^
CRRT (*n*, %)	10 (13)	9 (36)
ICUAW (*n*, %)	10 (13)	8 (32)
Medication in the ICU		
Steroid treatment in the ICU (*n*, %)	13 (18)	6 (29)

^1^ missing values in 5 and 4 of the patients, respectively; ^2^ missing values in 34 and 8 of the patients. Abbreviations: Body Mass Index (BMI), SAPS3, Simplified Acute Physiology Score 3; LPC, leukocyte particle concentration; ICU, intensive care unit; ACEi, angiotensin-converting enzyme inhibitor; ARB, angiotensin receptor blocker; PaO_2_/FiO_2_ ratio, arterial oxygen partial pressure-to-fractional inspired oxygen ratio; CRRT, continuous renal replacement therapy; ICUAW, ICU-acquired weakness.

**Table 2 biomedicines-10-00934-t002:** The evolution of surface and intracytoplasmic leukocyte marker levels in the percent of total leukocytes in ICU patients and in healthy controls (HCs). The median was the interquartile range. Leukocytes were defined as positive for CD45. Absolute LPC was in the order of 10^9^ cells per liter. Superscript numbers represent the number of missing values.

FCM	Healthy	ICU Patients					
Marker	Controls	Week 1	Week 2	Week 3	Week 4	Week 5	Week 6
	*n* = 12	*n* = 23	*n* = 10	*n* = 9	*n* = 6	*n* = 5	*n* = 3
CD2+	26 (18–31)	6 (4–10) **	2 (2–3) ^2^	3 (2–7) ^2^	5 (5–9)	2 (1–4)	4 (4–9)
CD4+	15 (11–18)	2 (1–4) **	1 (1–3)	1 (1–4)	3 (2–5)	0 (0–1)	3 (2–6)
CD8+	5 (3–9)	1 (1–2) **	1 (0–2)	1 (0–1)	1	1 (0–1)	1 (1–2)
CD158d+	23 (20–31)	4 (2–6) **	3 (2–4)	3 (2–6)	4 (3–6)	2 (0–2)	4 (4–9)
CD25+	29 (19–31)	7 (4–13) **	5 (4–9)	6 (5–10)	8 (6–13)	2 (2–5)	7 (7–14)
CD127+	20 (17–26)	5 (3–7) **^, 13^	2 (2–3) ^4^	2 (2–4) ^3^	5 (3–8)	4 (3–9)^2^	4 (2–8)
CD19+	2 (1–3)	0 (0–1) **	1 (0–2)	1	1 (0–1)	0 (0–1)	1 (0–1)
CD14+	4 (1–6)	2 (1–2)	1 (1–3)	2 (2–5)	3 (2–4)	4 (3–5)	4 (4–5)
CD203c+	0 (0–6)	0	0	0	0	0	0
CD33+	75 (66–86)	91 (87–95) *^,6^	95 (92–96) ^2^	94 (91–96)	91 (90–94) ^2^	96 (83–97)	93 (87–95)
HLA-DR+	94 (90–99)	92 (87–96)	96 (90–98)	97 (95–98)	96 (93–98)	98 (97–99)	95 (90–96)
CD55+	98 (87–99)	95 (93–96) ^13^	95 (90–97) ^5^	93 (91–94) ^2^	95 (90–96)	93 (92–94) ^2^	92 (91–94) ^1^
CD59+	94 (88–97)	97 (95–98) ^15^	94 (93–96) ^5^	92 (90–96) ^2^	92 (87–95)	88 (85–93) ^2^	91 (89–93) ^1^
CD45+	90 (83–91)	97 (96–98)	96 (94–98)	97 (97–98)	97 (96–97)	98 (97–99)	98 (97–98)
LPC		7.7 (6.5–12.4)	13.1 (9.3–18.4)	13.6 (9.0–19.2)	15.1 (12.6–17.9)	15.9 (11.9–18.8)	14.6 (14.5–16.1)
	*n* = 12	*n* = 7	*n* = 6	*n* = 6	*n* = 4	*n* = 3	*n* = 2
cIL10+	9 (5–54)	1 (1–15)	8 (1–39)	6 (3–8)	6 (2–13)	34 (17–51) ^1^	7 (4–10)
cIL1r2+	100 (99–100)	61 (2–83) **	36 (8–69)	73 (59–95)	83 (69–88)	62 (43–81) ^1^	100
cIL6+	5 (3–33)	1 (0–13)	0 (0–12)	4 (2–6)	6 (2–11)	3 (2–4)	3 (3–4)
cIL8+	5 (3–33)	1 (0–6)	0 (0–1)	3 (3–4)	4 (3–5)	3 (2–5)	4 (3–5)
cTNFα+	99 (99–100)	9 (6–32) **	29 (11–42)	31 (21–54)	47 (34–66)	61 (42–80)	100

** *p* < 0.001 and * *p* < 0.05 by the Mann–Whitney U-test for patients during their first week of ICU admission compared to HC. Abbreviations: c, intracellular; IL, interleukin; cIL1R2, intracellular interleukin 1 receptor type 2; TNFα, tumor necrosis factor alpha.

**Table 3 biomedicines-10-00934-t003:** The Spearman’ rank correlation coefficients for arterial oxygen partial pressure-to-fractional inspired oxygen (PaO_2_/FiO_2_) ratios in kPa in the third column and Krea_max_ in mmol/L in the right column correlated with CD markers, intracellular (prefix c) and plasma cytokine. All values were obtained during the patients first week of ICU admission.

Variables	PaO_2_/FiO_2_	PaO_2_/FiO_2_	Krea_max_	Krea_max_
	*n*	Spearman R	*n*	Spearman R
CD2+	15	−0.053	20	0.27
CD4+	17	−0.045	23	0.025
CD8+	17	0.28	23	0.011
CD158d+	17	−0.059	23	−0.21
CD25+	17	−0.036	23	0.12
CD127+	8	−0.060	9	0.33
CD19+	17	0.12	23	−0.0059
CD14+	17	−0.30	23	−0.28
CD203c+	17	0.33	23	−0.14
CD33+	13	−0.40	17	−0.42
HLA-DR	17	−0.16	23	−0.20
CD55+	8	0.28	9	−0.07
CD59+	8	−0.20	9	−0.27
CD45+	17	0.15	23	−0.29
cIL10+	6	0.77	7	−0.071
cIL1r2+	6	0.086	7	−0.14
cIL6+	6	0.71	7	−0.036
cIL8+	6	0.66	7	−0.27
cTNFα	6	0.29	7	0.14
IL-2	16	−0.13	23	0.29
IL-4	16	−0.098	23	0.30
IL-6	16	−0.49	23	0.39
IL-8	16	−0.42	23	0.44 *
IL-10	16	0.066	23	0.29
GM-CSF	16	0.025	23	0.14
IFNγ	16	−0.14	23	0.096
TNFα	16	−0.20	23	0.51 *

* *p*-value < 0.05 by Spearman’s rank-order correlation coefficients.

**Table 4 biomedicines-10-00934-t004:** Morphologies from 16 ICU patients with aberrant signals in FCM.

	Week 1	Week 2	Week 3	Week 4	Week 5	Week 6	Week 7
*n* (Female, Male)	16 (3, 13)	7 (2, 5)	6 (2, 4)	6 (2, 4)	2 (0, 2)	2 (0, 2)	1 (0, 1)
**Morphology**							
Thrombocyte anicystosis	16 (3, 13)	7 (2, 5)	6 (2, 4)	6 (2, 4)	2 (0, 2)	2 (0, 2)	1
Granulocyte							
Hypersegmentation	11 (3, 8)	4 (2, 2)	3 (2, 1)	4 (2, 2)	2 (0, 2)	1 (0, 1)	1
Hypergranulation	4 (0, 4)	3 (1, 2)	3 (2, 1)	0	0	0	1
Pelger cells	15 (3, 12)	6 (2, 4)	6 (2, 4)	5 (2, 3)	1 (1, 0)	1 (0, 1)	0
Hypogranulation	4 (1, 3)	5 (2, 3)	1 (0, 1)	4 (2, 2)	1 (1, 0)	0	0
Erythrocyte anicystosis	4 (1,3)	2 (0, 2)	1 (0, 1)	1 (1, 0)	1 (1, 0)	1 (0, 1)	0
Progenitor cells	5 (1, 4)	5 (1, 4)	4 (1, 3)	1 (1, 0)	0	1 (0, 1)	0

## Data Availability

Data are available from the corresponding author on reasonable request (https://doi.org/10.17044/scilifelab.14229410, accessed on 18 March 2021).

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
