# Peer review of "The Evolution of Blood Cell Phenotypes, Intracellular and Plasma Cytokines and Morphological Changes in Critically Ill COVID-19 Patients"

_biomedicines, 2022, doi:10.3390/biomedicines10050934_

Round 1

Reviewer 1 Report

In the manuscript from Berghall and colleagues overview the expression of several inflammatory and immune response markers in COVID-19 patients analyzed over time, emphasizing the different expression of CD33+ on leukocytes and the circulating IL6 in patients with different respiratory support. It also highlighted important changes in neutrophils morphology (dysplasia).

Despite being an interesting work, the results are not well described and there are misconceptions about the methodology that need to be considered in order to have more correct analyses.

In all cytometry analysis tables lack a description of the unit of measurement, which should be percentages of positive cells in relation to the total leukocyte population.

This is an approach that is not used in Flow cytometry analyses, in which the percentages refer to well-defined subpopulations: e.g. in leukocytes (CD45+) the T lymphocytes are CD3+ subpopulations in which CD4+ and CD8+ are analyzed, in B lymphocytes the CD19+.

Of course, the percentages of positive cells using the gating strategies change significantly. The authors should perform analyses using this approach to confirm the significant differences observed previously. It would also serve to graph the gating strategies used (as supplementary data). In this way, HLA DR and CD25 activation markers, for example, which do not result significant, could yield interesting results by analyzing their expression in different subpopulations.

For the results on CD33 and IL6, (lines 190-192) it is stated that in patients with PaO2/FiO2 less than 13, these markers are higher than in the group with less respiratory distress but this result does not refer to any figure and only correlations are reported.

Also for the analysis with respect to gender, it would be necessary to refer to the new gating strategies suggested above.

Finally, it would be interesting to evaluate whether compared to other clinical data the new phenotypic analysis could bring new statistically significant information.

In conclusion, the work, although interesting, needs a major revision in order to be evaluated for publication.

Author Response

We would like to thank the reviewers for their insightful comments on the manuscript, which has been substantially improved by the suggested revisions. We hope that these changes and answers have addressed the reviewers concerns. Please find the point-by-point answers below.

Best regards,

Elisabeth Berghäll

Reviewer 1

In the manuscript from Berghall and colleagues overview the expression of several inflammatory and immune response markers in COVID-19 patients analyzed over time, emphasizing the different expression of CD33+ on leukocytes and the circulating IL6 in patients with different respiratory support. It also highlighted important changes in neutrophils morphology (dysplasia).

Despite being an interesting work, the results are not well described and there are misconceptions about the methodology that need to be considered in order to have more correct analyses.

In all cytometry analysis tables lack a description of the unit of measurement, which should be percentages of positive cells in relation to the total leukocyte population.

We thank the reviewer for noticing this omission. Units have now been added to all the tables.

This is an approach that is not used in Flow cytometry analyses, in which the percentages refer to well-defined subpopulations: e.g. in leukocytes (CD45+) the T lymphocytes are CD3+ subpopulations in which CD4+ and CD8+ are analyzed, in B lymphocytes the CD19+.

CD45 is used as a leukocytes positive marker. This is so that thrombocytes, debris, dead cells can be omitted. Many different markers can be used to identify different subpopulations, CD3 identifies T-lymphocytes as well as CD2, CD5, CD7 etc. the amount of CD4+CD8=CD3 for instance. As for B-lymphocytes, CD19, CD20 among others are widely used.

Of course, the percentages of positive cells using the gating strategies change significantly. The authors should perform analyses using this approach to confirm the significant differences observed previously. It would also serve to graph the gating strategies used (as supplementary data). In this way, HLA DR and CD25 activation markers, for example, which do not result significant, could yield interesting results by analyzing their expression in different subpopulations.

All gating are made using a negative control to set the gate.  The negative control used is of unstained cells as well as isotypes for Immunoglobulins ie IgG1, IgG2a, b etc. Example added in Supplements Figure 1. This approach is used for all the markers and is according to routine procedures in the lab, which is accredited for the method as well as the interpretation of the flowcytometry plots. All populations have been analyzed according to cell lineage as well as aberrant phenotypes. This method is used widely in all routine flowcytometry labs.

For the results on CD33 and IL6, (lines 190-192) it is stated that in patients with PaO2/FiO2 less than 13, these markers are higher than in the group with less respiratory distress but this result does not refer to any figure and only correlations are reported.

The text in results, section 3.4, has been modified, mean±SD has been added for the group of patient with low PaO2/FiO2-ratio vs high ratio for CD33+ leukocytes (89.9±7.8% vs 84.6± 2.2%) and for IL-6 (234.0±518.9 vs. 2.3±1.7pg/ml) respectively. There was no correlation between PaO2/FiO2-ratio and the measured variables, but since there was a small difference at group level it was worth mentioning as an idea for larger studies. We have also added that these results should be interpreted with caution due to small subgroups with different organ failure in limitations in the discussion, section 4.

Also for the analysis with respect to gender, it would be necessary to refer to the new gating strategies suggested above.

Gating has been performed using accredited methods ie negative controls such as isotypes for Ig and unstained patient samples. All populations have been analyzed for aberrant phenotypes, please see earlier comment. We have omitted the gender aspect.

Finally, it would be interesting to evaluate whether compared to other clinical data the new phenotypic analysis could bring new statistically significant information.

Gating has been performed using accredited methods ie negative controls such as Ig and unstained patient samples. An example is added in Supplementary Figure 2. All populations have been analyzed for aberrant phenotypes. No further aberrant phenotypes were detected using the accredited, certified analysis methods used.

In conclusion, the work, although interesting, needs a major revision in order to be evaluated for publication.

We hope that we have been able to adequately address the reviewers’ questions here and in the revised manuscript.

Reviewer 2 Report

This study is well structured. The methods used are appropriate. The results are clear and useful. More patients are needed for reliable confirmation.

Author Response

We would like to thank the reviewers for their insightful comments on the manuscript, which has been substantially improved by the suggested revisions. We hope that these changes and answers have addressed the reviewers concerns. Please find the point-by-point answers below.

Best regards,

Elisabeth Berghäll

Reviewer 2

This study is well structured. The methods used are appropriate. The results are clear and useful. More patients are needed for reliable confirmation.

Thank you very much. We agree that more patients would be preferable. However, the logistics of fresh sample preparation limited the analysis capacity. Importantly, patients were included during a time frame when chloroquine and other experimental treatments had ceased, but before advanced immune-modulating treatment was introduced. This was ambiguously worded in the previous manuscript and has been clarified in materials and methods, section 2. Therefore, this represents a unique population of the natural course of illness for COVID-19, and we regret to say that we have not been able to find another site where the same analysis has been performed. Additional comments has been added in discussion, section 4.

Reviewer 3 Report

The paper by Dr. Bergfall applied FCM methods to analyze COVID-19 patients. As authors described few study demonstrated FCM results on COVIDpateints.  The results clearly demonstrated the resuction of lymphocyes in COVID-19.   To imrpove the manscripts the following desctriprions should be given.

Major;

The purpose of FCM study for COVID-19 should be more cleary presented.

Regarding the patients, all  patients were tretaed with chloroquine before sstudy.  The effect of the theapy on leukocyte subsets should also be considered.  It should be described only respiratory aid is given at ICU.

The methods for FCM should also be described. How you can alayze various CD antigens using relatively small amount of blood. Accordingly, the charcter of numbers given in Table 3 and 4 should be described.

Minor

The reason why HLA-DR+ dose not decrease should be given. Aberrant expression of CD158d in granulocte can not be explained by increases expression of Fc receptors. How do you block FC receptors for staining?

How fold diluted plasma was uded  for luminex analysis?

Author Response

We would like to thank the reviewers for their insightful comments on the manuscript, which has been substantially improved by the suggested revisions. We hope that these changes and answers have addressed the reviewers concerns. Please find the point-by-point answers below.

Best regards,

Elisabeth Berghäll

Reviewer 3

The paper by Dr. Berghall applied FCM methods to analyze COVID-19 patients. As authors described few study demonstrated FCM results on COVID patients.  The results clearly demonstrated the reduction of lymphocytes in COVID-19.   To improve the manuscripts the following descriptions should be given.

Major;

The purpose of FCM study for COVID-19 should be more Cleary presented.

The purpose of using FCM was to see the amount of cells positive or negative for the markers used in different cell lineages and also to see if there were any aberrant phenotypes present that maybe could be explained by the SARS-CoV2 virus. This has now been clarified in the paper.

Regarding the patients, all patients were treated with chloroquine before study.  The effect of the therapy on leukocyte subsets should also be considered.  It should be described only respiratory aid is given at ICU.

None of the patients had been given chloroquine, the study was made after we had stopped giving chloroquine routinely to COVID-19 patients and before advanced immune-modulating therapy was introduced. Therefore our results would be as close to the natural immunological course of the severe disease of SARS-CoV-2 as possible. This was ambiguously expressed in the manuscript and has now been revised and explained in the materials and methods, section 2, comments are added in section 4, discussion.

The methods for FCM should also be described. How you can analyze various CD antigens using relatively small amount of blood. Accordingly, the character of numbers given in Table 3 and 4 should be described.

We can analyze several antigens at the same time using different fluorochrome. In routine we use 10-12 different fluorochrome in one tube. For this study we used four different fluorochrome depending on how many lasers the instrument has and at which wavelength these lasers can detect. That way analysis with very small amount of sample can be performed. The method for using flowcytometry has been described numerous times before and was performed according to manufacturers’ recommendations. The techniques used in this paper is from an accredited laboratory which uses these techniques in the routine work that has been assessed by external qualified officials. We have now added additional information about the method in the paper according to your suggestions in materials and methods, section 2.2.

Minor

The reason why HLA-DR+ dose not decrease should be given. Aberrant expression of CD158d in granulocyte cannot be explained by increases expression of Fc receptors. How do you block FC receptors for staining?

Regarding HLA-DR+ we measured on the whole CD45+ leukocyte population, HLA-DR should not decrease since it is expressed on leucocytes. We did not see any differences in HLA-DR expression. The studies mentioned in introduction and discussion measured HLA-DR on monocytes. Thus, there we compared HLA-DR expression in two different population, total CD45+ leukocytes and monocytes. It was not clearly explained in the previous manuscript, and has been clarified in results in section 3.4, legends below Table 2 and discussion in section 4.   

The method for preparing the samples for flowcytometry analysis has been made according to the laboratory routine. The CD158d expression on the granulocytes have not been seen earlier and is an aberrant expression probably due to the infection of the SARS-CoV-2 virus. The importance of this remains to be seen. The antibodies used are specific for each receptor. Remaining antibody that does not bind gets washed away. Samples were prepared according to manufacturers’ protocol.

How fold diluted plasma was used for luminex analysis?

The dilution rate for plasma samples for Luminex analysis was four, it has been added in materials and methods, section 2.3.

Round 2

Reviewer 1 Report

The authors have sufficiently addressed most of the issues raised in the first round of review.

Author Response

Dear Reviewer,

There were no questions from You this time so I have no furhter comments or answers. Thankyou for Your time and insightful comments!

Best regards Elisabeth Berghäll  

Reviewer 3 Report

The authors revised the paper and added methods for the analysis.

There are still major flaws.

The ethical committee clearance was not described.

The authors described the importance of the decline of CD127 in conclusion, however,

the data of CD127 is missing in Table 2.

The data are solely dependent on the percentages of leukocytes, absolute numbers of each parameter should be calculated followed by statistical analysis.  

The title of Table 3 is the same as that of Table 2.

Author Response

Please se attachment.

Round 3

Reviewer 3 Report

I couod not find out the answer to my previous question below.

"The data are solely dependent on the percentages of leukocytes, absolute numbers of each parameter should be calculated followed by statistical analysis".  I hope you can understand the measning of "absolute numbers"

Title of Table 2B

Which molecules are related with CD (cluster differentiation)?

Author Response

Dear Reviewer,

Thank you for your time and valuable time and insightful comments. We are sorry we didn’t address your questions properly in the last round and thus caused you extra work. Hopefully we now have answered your questions well enough, in this complex and fast developing field of flow cytometry. Please find the point-by-point answers below.

Best regards 
Elisabeth Berghäll

1. "The data are solely dependent on the percentages of leukocytes, absolute numbers of each parameter should be calculated followed by statistical analysis".  I hope you can understand the measning of "absolute numbers"

       Flow cytometry results are often given as a percentage and the statistical analyzes are performed accordingly. The statistics do not differ when using absolute numbers.

2. Title of Table 2B 
       Regarding title of Table 2B I don’t fully understand the question, but have made minor changes, marked in green.

3. Which molecules are related with CD (cluster differentiation)?

       We believe that those who read this are acquainted with the CD-marker concept and the molecules that the CD markers stand for. Some are briefly described in the introduction, others are well known. Their exact and full characteristics are often complex and can easily be looked up. We have attached the first column of description that we used as working material, to the requested ABS-table. However, the descriptions are not scientifically complete and concise and therefore we think it is not suitable for publishing. But we are of course happy to share it with reviewers and editor.

Round 4

Reviewer 3 Report

Absolute  numbers could be calculated as  below.

Absolute numbers =Percent x numbers of leukocytes. I hope you could  calculate and analyze based on the absolute neumbers.

I do not think any molecules in table 2B could be expressed as CD molecules. CD markers are usually against cell surafce markers.

Cytokines do not have CD numbers. It should be expressed "Intra cytoplasmic ytokines" 

Author Response

Dear Reviewer

Thankyou for your patience. The manuscript with Table 2B has now been modified in order to be correct regarding CD and cytokine terminology. 

Best regards Elisabeth Berghäll